# Sleep Should Be Focused on When Analyzing Physical Activity in Hospitalized Older Adults after Trunk and Lower Extremity Fractures—A Pilot Study

**DOI:** 10.3390/healthcare10081429

**Published:** 2022-07-30

**Authors:** Yoichi Kaizu, Takeaki Kasuga, Yu Takahashi, Tomohiro Otani, Kazuhiro Miyata

**Affiliations:** 1Department of Rehabilitation Center, Hidaka Hospital, 886 Nakao-machi, Takasaki, Gunma 370-0001, Japan; kaiduyoichi@yahoo.co.jp (Y.K.); t.kasemp-26.hrnzm@outlook.com (Y.T.); 2Department of Rehabilitation Center, Hidaka Rehabilitation Hospital, 2204 Yoshii-machi Maniwa, Takasaki, Gunma 370-2104, Japan; tkhsyy@gmail.com; 3Department of Physical Therapy, Ota College of Medical Technology, 1373 Higashinagaoka-cho, Ota, Gunma 373-0812, Japan; ta-to-mo.20329@sirius.ocn.ne.jp; 4Department of Physical Therapy, Ibaraki Prefectural University of Health Science, 4669-2 Ami-Machi, Inashiki-gun, Ibaraki 300-0394, Japan

**Keywords:** older adults, sedentary behavior, sleep, physical activity, inpatient, fracture

## Abstract

Although the importance of resting in bed for hospitalized older adults is known, current methods of interpreting physical activity (PA) recommend the use of a broad definition of sedentary behavior (SB) that includes 0–1.5 metabolic equivalents (METs) of sleep (SL) and sitting. We investigated the characteristics of PA by conducting a cross-sectional study of 25 older adults with trunk and lower extremity fractures. The intensity of their PA was interpreted as SL (0–0.9 METs), SB (1–1.5 METs), low-intensity PA (LIPA: 1.6–2.9 METs), and moderate-to-vigorous PA (MVPA: >3.0 METs). We calculated the correlation coefficients to clarify the relationship between each PA intensity level. Our analyses revealed that the PA time (min/day) was accounted for by SB (53.5%), SL (23.2%), LIPA (22.8%), and MVPA (0.5%). We observed negative correlations between SL and SB (r  =  −0.837) and between SL and LIPA (r  =  −0.705), and positive correlations between SB and LIPA (r  =  0.346) and between LIPA and MVPA (r  =  0.429). SL and SB were also found to have different trends in relation to physical function. These results indicate that SL and SB are trade-offs for PA during the day. Separate interpretations of the SL and SB of older hospitalized adults are thus recommended.

## 1. Introduction

Fragility fractures are a serious problem in the older and very old. For example, in Asia, hip fracture incidence is projected to increase from 1.12 million in 2018 to 2.56 million (2.28 times) by 2050 [1]. The incidence of vertebral fractures is reported to be 10 times higher than that of hip fractures [2], which is also a problem. Furthermore, older adults hospitalized following a fracture are at risk for hospitalization-associated disability caused by low levels of physical activity (PA) [3]. Hospitalized-associated disability causes impaired independence in the daily lives of hospitalized older adults. Among fragility fractures, lower extremity fractures are more likely to have lower levels of physical activity than upper extremity fractures [4,5]. Furthermore, trunk fractures (vertebral fracture or pelvic fracture) are also more likely to have low PA levels due to early hospitalization treatment with rigorous bed rest [6] and load restriction in the early stages [7]. Therefore, appropriate monitoring of PA levels in patients after trunk and lower extremity fractures during hospitalization is important.

For measuring the amount of physical activity, the use of objective data obtained by an accelerometer is recommended, based on this device’s high reliability and validity [8]. When classifying the intensity of physical activity using metabolic equivalents (METs), a consensus has been achieved to use the following definitions: sedentary behavior (SB; 0–1.5 METs), low-intensity physical activity (LIPA; 1.6–2.9 METs), and moderate-to-vigorous physical activity (MVPA; >3.0 METs) [9]. Most of the studies using METs, including those during hospitalization, have used these definitions. However, studies that used devices and defined physical activity using methods other than METs [10,11] and behavioral mapping methods [12] indicated that hospitalized older adults spend 57–83% of their time lying in bed. Although this finding suggests the importance of the time spent resting in bed for hospitalized older adults, the recommended definition of physical activity intensity is more broadly defined as SB (0–1.5 METs), combined with sleep (SL; 0–0.9 METs) and narrowly defined SB (1–1.5 METs). Hereinafter, 1–1.5 METs will be referred to as “SB,” and 0–1.5 METs will be referred to as “broadly defined SB” in this paper.

According to the results of measurements using self-reporting [13] and non-METs devices in community-dwelling older adults [14], the daytime lying-down time ranged from 7.7% to 17.4%, which is less than the corresponding values reported in hospitalized older adults. Thus, the amount of SL time is low among community-dwelling older adults. The use of broadly defined SB may be less problematic, but SL is likely to be more important during hospitalization. Whether or not SL should be adopted as the METs definition in studies of physical activity in hospitalized older adults is an important issue. The importance of not only the total time but also the duration (bout) of each physical activity intensity level has also been described [15,16], but the details of each activity intensity during the hospitalization of older adults who have experienced trunk or lower extremity fractures are unknown. Furthermore, trends in the relationship between PA and physical function (walking speed and balance function) are also important. Previous studies have found that broadly defined SB is negatively associated with physical function [17], while PA (LIPA, MVPA) is positively associated [18]. However, SL and SB may possibly have different trends in their association with physical function. Since PA plays an important role as a predictor of physical function, SL and SB should be analyzed separately if they have different trends in association with physical function.

We conducted the present study to clarify the distribution of physical activities, including SL, and to assess the relationships among the SL, SB, LIPA, and MVPA of hospitalized older adults with trunk and lower extremity fractures. A secondary objective was to clarify the details of the intensity of each physical activity and the relationship between physical activity and physical function. Using this information as a basis for decision-making, we also consider recommendations regarding whether SL and SB should be determined together or separately in hospitalized older adults.

## 2. Materials and Methods

### 2.1. Participants

With the approval of the Hidaka Hospital Ethics Committee (approval no. 285), we conducted a cross-sectional study of adults with a trunk (vertebral fracture or pelvic fracture) or lower extremity fracture who were admitted to the convalescent ward of Hidaka Hospital between March 2019 and June 2021. The inclusion criteria were age ≥ 65 years, having suffered a fracture of the trunk or a lower extremity, having exceeded the cutoff value of the dementia screening test (Hasegawa Dementia Scale revised version (Japanese ver.) ≥ 21 [19]), and having the ability to walk independently. Patients with a history of stroke or with missing data were excluded. Each patient provided written informed consent prior to recruitment.

The required sample size is 20, calculated with the expected correlation coefficient value of 0.6 (moderate strong), alpha = 0.05, and power = 0.8.

### 2.2. Data Collection

The patients’ basic information, discharge parameters, physical function, and physical activity were measured. The basic information included age, gender, body mass index (BMI), and type of fracture. The parameters at discharge were destination (home or facility), length of hospital stay (number of days from surgery to discharge), and functional independence measure (FIM).

### 2.3. Physical Activity

To evaluate the physical activity of the patients after they achieved gait independence, we used a three-axis accelerometer (model HJA-750C, Omron, Kyoto, Japan) to measure the METs of the patient every 60 sec for 12 h from 7:00 to 19:00 over 3 consecutive days [20,21]. The measurement time of 7:00 to 19:00 was defined by the following conditions: (1) the time of day when they are not usually lying in bed and (2) when they are considered to be eligible for physical activity promotion. HJA-750C is a compact (40 mm × 52 mm × 12 mm) and lightweight (23 g) three-axis accelerometer that estimates METs. The device records forward/backward (*x*-axis), left/right (*y*-axis), and up/down (*z*-axis) accelerations, with a resolution of 3 mG and a sampling rate of 32 Hz. Each of the three signals from the tri-axis accelerometer was passed through a high-pass filter with a cutoff frequency of 0.7 Hz to remove the gravity acceleration component from the signals. Acceleration signals are integrated over a 10 s epoch. The MET estimation algorithm, validated using the Douglas Bag method, categorizes PA into exercise, housework, and sitting, and applies a specific linear regression model to each activity [22,23]. Since 3–5 days of measurement is necessary to obtain high reliability in measuring physical activity using accelerometers, 3 days of measurement were employed in this study [20,24]. The validity study of this physical activity meter showed a significant correlation between actual and predicted METs for both older (r = 0.85, *p* < 0.001) and younger (r = 0.88, *p* < 0.001) subjects [23].

The accelerometer was attached to the patient’s lower back, and the total activity time and bouts were measured for four activity intensities: 0–0.9 METs as SL [25], 1–1.5 METs as SB, 1.6–2.9 METs as LIPA, and >3.0 METs as MVPA [9]. Note that hospitalized older adults have been shown to spend 57%–83% of their time lying in bed [10,11,12], suggesting the importance of the time spent lying in bed by hospitalized older adults. Therefore, in this study, broadly defined SB (0–1.5 METs) was analyzed separately for SL and SB.

Bouts were calculated for SL (<30 min, 30–59 min, 60–89 min, and >90 min), SB (<30 min, 30–59 min, 60–89 min, and >90 min), and T-PA (LIPA + MVPA; 0–1 min, 2–4 min, and >5 min) as a percentage of the daily activity time, and then averaged by the number of effective days [15].

### 2.4. Physical Function

The measurements of physical function were the Berg Balance Scale (BBS) and the Timed Up and Go (TUG) test at the time point at which the patient achieved gait independence. The BBS consists of 14 items of activities of daily living and can comprehensively assess not only balance ability but also physical functions [26]. The BBS has been recognized for its reliability and validity as an evaluation scale [27,28]. The TUG test requires subjects to rise from a standard armchair, walk to a marker three meters away, turn, return, and sit down again. This test is a reliable and valid test for quantifying functional mobility [29].

### 2.5. Statistical Analysis

For normally distributed continuous variables, the standard deviation and mean were computed. If data were not normally distributed, the median and interquartile range (25th–75th percentile) were computed. Categorical variables are indicated by the number of people. To clarify the relationships among the time spent (i.e., time volume) in each activity-intensity (SL, SB, LIPA, MVPA), and to evaluate the relationship between each activity intensity’s time volume and the patients’ age and physical function indices (BBS, TUG), we calculated correlation coefficients (Pearson for indices with normal distribution, and Spearman for indices without). The strength of the correlation was defined as a correlation coefficient (r) of <0.3 (−0.3) as no correlation, 0.3 to 0.5 (−0.3 to −0.5) as fair, 0.6 to 0.8 (−0.6 to −0.8) as moderately strong, and >0.8 (> −0.8) as very strong [30].

The interpretation of the correlation between activity intensities is that a negative correlation implies a trade-off in intraday PA, in which case the two should be analyzed separately. For example, if SL is not found to be related to other activity intensity times, it means that the amount of time spent on SL is approximately constant across subjects. In that case, SL could be interpreted as an indicator that does not need to be analyzed separately. Alternatively, if SL and other activity intensity times are positively correlated, this means that they have a similar trend, and in this case, too, SL can be interpreted as an indicator that does not need to be analyzed separately. Of particular interest to us is the association between SL and SB. A negative correlation between SL and SB implies 1) that the amount of time spent on SL differs between subjects and 2) that a trade-off relationship exists between the amount of time spent on SB. In such cases, SL and SB are considered better indicators to be analyzed separately. As an interpretation of the relationship between physical function indices and each activity intensity, if SL and SB have correlation coefficients with different trends, they are likely to act differently as physical function-related factors and are considered indicators that should be analyzed separately.

Statistical significance was based on two-sided *p*-values of <0.05. All statistical analyses were performed using SPSS statistical software ver. 26 (IBM, Armonk, NY, USA).

## 3. Results

Thirty-six patients met the inclusion criteria; of them, five patients did not agree to participate in the study, five had had a previous stroke, and one was excluded due to missing data. A final total of 25 patients was thus analyzed. Table 1 summarizes the patients’ basic information, discharge parameters, physical function at the time of the measurement of physical activity, and physical activity. The patients were relatively older (age 79.4 ± 7.5), included more females (4 males, 21 females), had large numbers of vertebral compression fractures (*n* = 14) and hip fractures (*n* = 7), and all were returning home (*n* = 25).

The amount of time (min/day) for each physical activity intensity was dominated by SB (53.5%), SL (23.2%), LIPA (22.8%), and MVPA (0.5%). The bouts of each physical activity intensity are illustrated in Figure 1. The patients spent 41.3 min per day on SL, which continued for >30 min per day, accounting for 24.8% of all SL. Regarding the patients’ SB, SB that continued for >30 min per day accounted for 66.6 min per day, or 17.3% of all SB. As for T-PA, 94.1 min per day of T-PA was continued for >5 min per day, accounting for 56.3% of all T-PA.

The relationships among the physical activity intensity levels are shown in Table 2, and Figure 2 provides the scatter plots of the relationships between SL and SB and between SL and LIPA. The correlation between SL and SB was very strong (r  =  −0.837, *p* < 0.001, Figure 2a). The correlation between SL and LIPA was moderately strong (r  =  −0.705, *p* < 0.001, Figure 2b). The correlation between LIPA and MVPA (r  =  0.429, *p* = 0.032) was fair. The patients’ SB and LIPA showed a non-significant association (r  =  0.346, *p* = 0.091).

The relationship between each physical activity intensity’s time volume and physical function is shown in Table 3. A fair correlation was found between the patients’ SB and their scores on the BBS (r  =  −0.432, *p* = 0.031). Although not significantly associated, SL and the BBS (r  =  0.354, *p* = 0.083) were observed to be inversely related to SB and the BBS (r  =  −0.432, *p* = 0.031). Each of the physical activity intensity levels’ time volume and the TUG test results were not significantly related.

## 4. Discussion

We examined the association between the amount of physical activity, including SL, in hospitalized older adults who had experienced a trunk or lower extremity fracture. The results of our analyses revealed that the patients’ SL had a very strong negative correlation with their SB and a moderately strong negative correlation with their LIPA, whereas their SB and LIPA had a fair positive correlation. These results suggest that SL and SB have a trade-off relationship with respect to the amount of time for physical activity during the day, and that SB and LIPA may have a similar relationship. We also observed that the patients’ SL and SB showed an inverse relationship in relation to their BBS scores. These results suggest that SL and SB have an antagonistic rather than a similar relationship, and, thus, separate analyses of SL and SB—rather than broadly defined SB—are recommended in studies of physical activity in hospitalized older adults.

Our present findings demonstrated that SB–LIPA had positive correlation coefficients in hospitalized older adults, indicating that they are variables with proportional relationships rather than trade-offs. In addition, SL and SB (which have rarely been studied in community-dwelling older adults) showed a very strong negative correlation in our series of older patients, suggesting the possibility of a trade-off relationship between SL and SB. In the past, the trade-off structure of broadly defined SB versus LIPA and MVPA has been discussed in studies of physical activity mainly in community-dwelling older adults, and the replacement of a large amount of SB time with LIPA and MVPA was proposed to improve health [31,32]. In a covariance study of community-dwelling older adults, SL and SB were shown to have the highest covariance, SB–LIPA and SB–MVPA had the lowest covariance, and SB was easily replaced with LIPA and MVPA [33]. Thus, the present results suggest differences in the temporal distribution of each physical activity intensity over the course of a day between hospitalized older adults and community-dwelling older adults.

Our present results indicate that in hospitalized older adults, SL and SB may have a conflicting relationship with the BBS score. The relationship that we observed between SB and physical function is similar to the relationship between physical function and broadly defined SB among community-dwelling older adults. On the other hand, the relationship between the amount of SL time and BBS is incongruous. Our present results are inconsistent with reports that prolonged bed rest leads to hospitalization-related dysfunction [5,6]. Since this pilot study was a cross-sectional univariate analysis, the effects of baseline differences and other confounding factors could not be excluded, and a longitudinal study or multiple analyses are thus necessary to clarify the causal relationships in this study. What is important to note here is that SL and SB may have different trends in their relationships with physical function.

The results of our present analyses demonstrate that SL accounted for 23.2% of hospitalized older adults’ daily physical activity, which is a larger proportion than that of community-dwelling older adults [13,14]. Our correlation analyses revealed a trade-off relationship between SL and SB, and SL and SB showed different trends in relation to physical function. We recommend that SL and SB be interpreted separately in hospitalized older adults because of (1) the trade-off between SL versus SB as the amount of time for physical activity, and (2) the inverse relationship for physical function. In the consensus project regarding the definition of SB [9], three respondents suggested that an interval of 1–1.5 METs should be used instead of ≤ 1.5 METs, and we speculate that this assertion may be correct, especially in hospitalized older adults.

In a study using broadly defined SB in hospitalized older adults after lower extremity fracture, the patients reported >10 h out of 13 h during the day (76.9%) [34], which is almost the same as the SL + SB value (77%) in the present study. Our search of the literature identified only one study that calculated the bouts for each physical -activity intensity level in hospitalized older adults: Miller et al.’s study of ambulatory independent patients after leg amputation [15]. In that study, the authors measured broadly defined SB. We, thus, cannot simply compare their results with ours, but we observed that the values of both SL + SB and T-PA were generally similar to the results of the Miller et al. study. On the other hand, the daily T-PA in hospitalized older hip fracture patients was reported to be 16–52 min [35], and the 163.9 min of T-PA in our present study is much larger. We suspect that this discrepancy is due to the fact that Miller et al. and we both limited the measurement time to after the patients achieved gait independence. When focusing on bouts, Miller et al. observed that the percentage of bouts of >5 min among total T-PA bouts was low (15.3%), whereas in the present study the corresponding value was 56.7%. We believe that disease specificity is likely to be involved in this difference in values. Overall, we observed that in hospitalized older adults after a trunk/lower extremity fracture, the broadly defined SB time was similar to that of previous studies, the SL/SB bout was relatively short, the total T-PA time was high, and the T-PA bouts were found to have a large percentage of dosing times >5 min.

This pilot study has several methodological limitations. The analysis of the association between physical activity and physical function was limited to a univariate analysis. Multiple confounding factors, such as age, are involved in the association between physical activity and physical function, and multiple analyses are needed to reveal independent associations. The fact that our patients were predominantly female and skewed toward older ages may also have affected the results of the analyses. The sample size of this study (*n* = 25), while meeting the target sample size indicated in the Materials and Methods section, is small compared to other studies on physical activity. Therefore, a larger study is needed in the future. In addition, we were not able to conduct an analysis that included the amount of activity before the patients’ hospitalization. Because this was a cross-sectional study, it is difficult to account for the differences in baseline values and infer causality. Future research should include a longitudinal study of the amount of physical activity during hospitalization, and analyses of the degree of changes in physical activity and factors.

## 5. Conclusions

The results of this study showed that hospitalized older adults with a trunk or lower extremity fracture spent nearly a quarter of their daytime physical activity SL, with a trade-off between SL and SB. Additionally, we found that the broadly defined SB time was similar to previous studies, SL/SB bout was relatively short, total T-PA bout was long, and T-PA bout had a large proportion of dose duration >5 min. SL and SB also showed different trends in their association with physical function. Based on these results, we recommend that SL and SB should be interpreted separately in hospitalized older adults.

## Figures and Tables

**Figure 1 healthcare-10-01429-f001:**
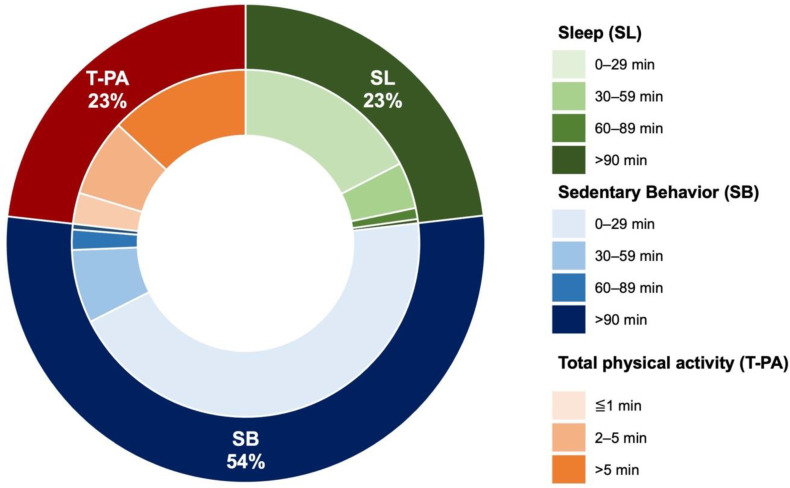
Percentage of time spent in sleep, sedentary behavior, and total physical activity (low-intensity physical activity + moderate vigorous physical activity) (outer ring). Proportion of time spent in each intensity of physical activity (inner ring; darker color indicates longer duration).

**Figure 2 healthcare-10-01429-f002:**
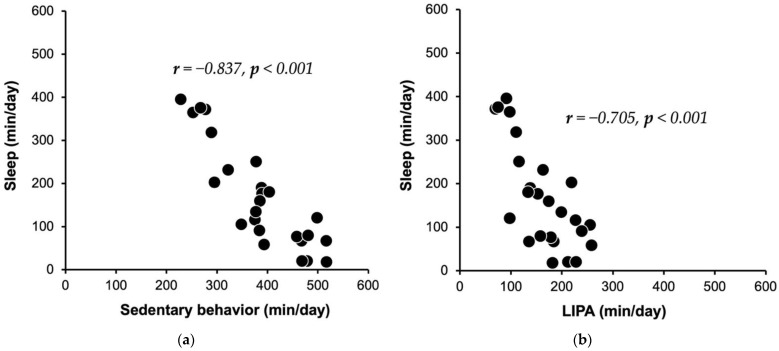
Scatter plots showing the associations for each physical activity intensity, showing data from 25 subjects. Each marker corresponds to an individual subject and is commonly used in plots (**a**,**b**). LIPA: low-intensity physical activity.

**Table 1 healthcare-10-01429-t001:** Basic information, discharge parameters, physical function, physical activity for all participants.

Variable		All Participants (*n* = 25)
** *Basic information* **		
Age, years		79.4 ± 7.5 (80, 73–85)
Sex, male/female		4/21
Body mass index, kg/m^2^		21.7 ± 3.3 (21.3, 20.4–23.1)
Type of fracture, pelvis fracture/vertebral compression fracture/hip fracture/distal femoral fracture/tibial plateau fracture		3/14/7/1
** *Discharge parameters* **		
Destination, home/facility		25/0
Length of hospital stay, days		52.8 ± 18.9 (49, 44–63)
Functional Independence Measure	Motor score	82.3 ± 7.6 (85, 77–89)
	Cognitive score	33.3 ± 2.5 (34, 32–35)
	Total score	115.6 ± 9.3 (120, 109–124)
** *Physical function at gait independence* **		
Berg balance scale		50 ± 5.2 (51, 46–55)
Timed Up and Go test, s		14.6 ± 6.8 (12.9, 9–17.5)
** *Physical activity* **		
SL (0–0.9 METs), min/day		167.5 ± 119.2 (134, 72–241)
SB (1–1.5 METs), min/day		385.8 ± 85.5 (385, 309–468)
LIPA (1.6–2.9 METs), min/day		163.9 ± 56.7 (163, 113–216)
MVPA (>3.0 METs), min/day		3.8 ± 3 (3, 2–5)

Values are shown as mean ± SD (median, interquartile range). SL: sleep, SB: sedentary behavior, LIPA: low-intensity physical activity, MVPA: moderate to vigorous-intensity physical activity, METs: metabolic equivalents.

**Table 2 healthcare-10-01429-t002:** Correlation between age and each physical activity intensity (*n* = 25).

	SL		SB		LIPA		MVPA	
	*r*	*p*	*r*	*p*	*r*	*p*	*r*	*p*
Age	−0.081	0.699	0.106	0.615	−0.017	0.935	0.24	0.91
SL			−0.837	<0.001	−0.705	<0.001	−0.183	0.381
SB					0.346	0.091	−0.031	0.885
LIPA							0.429	0.032
MVPA								

SL—sleep, SB—sedentary behavior, LIPA—low-intensity physical activity, MVPA—moderate-to-vigorous intensity physical activity.

**Table 3 healthcare-10-01429-t003:** Correlation between each physical activity intensity and physical function (*n* = 25).

	Berg Balance Scale		Timed up and Go Test	
	*r*	*p*	*r*	*p*
SL	0.354	0.083	−0.283	0.17
SB	−0.432	0.031	0.364	0.074
LIPA	−0.098	0.641	0.152	0.467
MVPA	0.036	0.864	−0.109	0.605

SL—sleep, SB—sedentary behavior, LIPA—low-intensity physical activity, MVPA—moderate-to-vigorous intensity physical activity.

## Data Availability

Not applicable.

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
