# Peer review of "Sleep Should Be Focused on When Analyzing Physical Activity in Hospitalized Older Adults after Trunk and Lower Extremity Fractures—A Pilot Study"

_healthcare, 2022, doi:10.3390/healthcare10081429_

Round 1

Reviewer 1 Report

Sleep should be focused on when analyzing physical activity in hospitalized older adults after trunk and lower extremity fractures - a pilot
study”(healthcare-1802193

1. Hasegawa Dementia Scale revised version 21 or higherWhat is the meaning of “higher”Did you mean the latest version of Hasegawa Dementia Scalesuch as version 22Did you use the original language version of Hasegawa Dementia Scale or translated version? this is the same case for Berg Balance Scale (BBS) Please clarify this and cite the related references. What is the standard of preserved cognitive function”?

2. Related references should be cited for Timed Up and Go (TUG) test.

3. Because physical function is also an important variable in the current investigationthus it had better to add a subtitle to introduce the measurement of physical function just as “2.3. Physical Activity”did. More details about Berg Balance Scale (BBS) and Timed Up and Go (TUG) test should be provided under physical function part to help the readers get better understanding of your results.

4. How did you determine your sample size? Did you calculate the sample size needed before formal study? 25 participants seems too little to get reliable results, especially for the correlation analysis. Also, little sample size should be mentioned in the limitation part.

Author Response

Manuscript healthcare-1802193: “Sleep should be focused on when analyzing physical activity in hospitalized older adults after trunk and lower extremity fractures - a pilot study” by Kaizu et al.

Reply to Reviewer #1

Major Points

  1. “Hasegawa Dementia Scale revised version 21 or higher”What is the meaning of “higher”?Did you mean the latest version of Hasegawa Dementia Scale,such as version 22?Did you use the original language version of Hasegawa Dementia Scale or translated version? (this is the same case for Berg Balance Scale (BBS) )Please clarify this and cite the related references. What is the standard of “preserved cognitive function”?
  2. Related references should be cited for Timed Up and Go (TUG) test.
  3. Because physical function is also an important variable in the current investigation,thus it had better to add a subtitle to introduce the measurement of physical function just as “3. Physical Activity”did. More details about Berg Balance Scale (BBS) and Timed Up and Go (TUG) test should be provided under “physical function” part to help the readers get better understanding of your results.

<Correction of HDS-R part> Added a reference to the HDS-R used, and revised the text to read as follows. In addition, the confusing phrase "retained cognitive function" has been revised to "exceeded the cutoff value of the dementia screening test".

(Line 88-)

“The inclusion criteria were age ≥ 65 years, having suffered a fracture of the trunk or a lower extremity, having exceeded the cutoff value of the dementia screening test (Hasegawa Dementia Scale revised version (Japanese ver.) ≥ 21 [19]), and having the ability to walk independently.”

  1. Kato, S. Development of the revised version of Hasegawa's Dementia Scale (HDS-R). Jpn Geriatr Psychiatr Med. 1991, 2, 1339-1347.

<Correction of BBS and TUG part> We added a section on "2.4. Physical Function" for BBS and TUG, and added the following explanation, citing the literature.

(Line 135-)

2.4. Physical Function

“The measurements of physical function were the Berg Balance Scale (BBS) and the Timed Up and Go (TUG) test at the time point at which the patient achieved gait in-dependence. The BBS consists of 14 items of activities of daily living and can comprehensively assess not only balance ability but also physical functions [27]. The BBS has been recognized for its reliability and validity as an evaluation scale [28,29]. The TUG test requires subjects to rise from a standard armchair, walk to a marker three meters away, turn, return, and sit down again. This test is a reliable and valid test for quantifying functional mobility [30].”

  1. Berg, K.; Wood-Dauphine, S.; Williams, J.I.; Gayton D. Measuring balance in the elderly: preliminary development of an instrument. Physiotherapy Canada. 1989, 41, 304-311.
  2. Berg, K.; Wood-Dauphine, S.; Williams, J.I. The Balance Scale: reliability assessment with elderly residents and patients with an acute stroke. Scand J Rehabil Med. 1995, 27, 27-36.
  3. Berg, K.; Wood-Dauphine, S.; Williams, J.I.; Maki, B. Measuring balance in the elderly: validation of an instrument. Can J Public Health. 1992, 83, S7-S11.
  4. Podsiadlo, D.; Richardson, S. The timed "Up & Go": a test of basic functional mobility for frail elderly persons. J Am Geriatr Soc. 1991, 39, 142-148. doi: 10.1111/j.15325415.1991.tb01616.x

  1. How did you determine your sample size? Did you calculate the sample size needed before formal study? 25 participants seems too little to get reliable results, especially for the correlation analysis. Also, little sample size should be mentioned in the limitation part.

The following statement regarding sample size calculation was added to the Methods, and the following statement was added to the Discussion section as a limitation of this study.

(Line 94-)

【Methods】“The required sample size is 20, calculated with the expected correlation coefficient value of 0.6 (moderate strong), alpha = 0.05, and power = 0.8.”

(Line 295-)

【Discussion】“The sample size of this study (n=25), while meeting the target sample size indicated in the Materials and Methods section, is small compared to other studies on physical activity. Therefore, a larger study is needed in the future.”

Reviewer 2 Report

please provides the validity or accuracy regarding the measurement (i.e., accelerometer); line 104-110. 

Number of the participants might be a problematic because of relatively small sample size.

Results: Table 1; MVPA is only 3.8 min/days. This is not be a MVPA. please provide the details and the limitation. 

Table 2: LIPA is related to MVPA (r=0.429, p=.032). please provide the results and some results are irrelevant for example SB and LIPA shown no significant. 

The association between SB and LIPA or SL and SB are interaction; please provide the details and that might lead to the results of the study. 

Discussion part: please explain the results again. 

Author Response

Reply to Reviewer #2

  1. Please provides the validity or accuracy regarding the measurement (i.e., accelerometer); line 104-110.

The following description and references were added to the Materials and Methods section.

(Line 119-)

Since 3-5 days of measurement is necessary to obtain high reliability in measuring physical activity using accelerometers, 3 days of measurement were employed in this study [24,25]. Validity study of this physical activity meter showed a significant correlation between actual and predicted METs for both older (r = 0.85, P < 0.001) and younger (r = 0.88, P < 0.001) subjects [23].

  1. Nagayoshi S.; Oshima Y.; Ando T.; Aoyama T.; Nakae S.; Usui C.; Kumagai S.; Tanaka S. Validity of estimating physical activity intensity using a triaxial accelerometer in healthy adults and older adults. BMJ Open Sport Exerc Med. 2019, 5, e000592. doi: 10.1136/bmjsem-2019-000592
  2. Kocherginsky, M.; Huisingh-Scheetz, M.; Dale, W.; Lauderdale, D.S.; Waite, L. Measuring Physical Activity with Hip Accelerometry among U.S. Older Adults: How Many Days Are Enough? PLoS One. 2017, 12, e0170082. doi: 10.1371/journal.pone.0170082
  3. Trost, S.G.; McIver, K.L.; Pate, R.R. Conducting accelerometer-based activity assessments in field-based research. Med Sci Sports Exerc. 2005, 37, S531-S543. doi: 10.1249/01.mss.0000185657.86065.98

  1. Number of the participants might be a problematic because of relatively small sample size.

The following statement regarding sample size calculation was added to the Methods, and the following statement was added to the Discussion section as a limitation of this study.

(Line 94-)

【Methods】“The required sample size is 20, calculated with the expected correlation coefficient value of 0.6 (moderate strong), alpha = 0.05, and power = 0.8.”

(Line 295-)

【Discussion】“The sample size of this study (n=25), while meeting the target sample size indicated in the Materials and Methods section, is small compared to other studies on physical activity. Therefore, a larger study is needed in the future.”

  1. Results: Table 1; MVPA is only 3.8 min/days. This is not be a MVPA. please provide the details and the limitation.

We were under the understanding that MVPA refers to exercise intensity (METs > 3.0) and as for duration, the WHO only "recommends that it be performed for at least 10 minutes". Therefore, we believe that even if the duration was less than 10 minutes, it would still qualify as MVPA.

  1. Table 2: LIPA is related to MVPA (r=0.429, p=.032). please provide the results and some results are irrelevant for example SB and LIPA shown no significant.

In the case of this study, as pointed out above, the MVPA was quantitatively small (3.8 min/day), so we did not consider it important to say that a correlation was found.

  1. The association between SB and LIPA or SL and SB are interaction; please provide the details and that might lead to the results of the study.

The details of how to interpret the correlation between each physical activity intensity are described in the Material and Methods section, and the interpretation of a positive correlation was added to the Methods section.

(Line 161)

“Alternatively, if SL and other activity intensity times are positively correlated, this means that they have a similar trend, and in this case, too, SL can be interpreted as an indicator that does not need to be analyzed separately.”

  1. Discussion part: please explain the results again.

The second and third paragraphs of the discussion have been changed to include the results of the current study at the beginning of the paragraphs and explain them.

(Line 236-)

“Our present findings demonstrated that SB-LIPA had positive correlation coefficients in hospitalized older adults, indicating that they are variables with proportional relationships rather than trade-offs. In addition, SL and SB (which have rarely been studied in community-dwelling older adults) showed a very strong negative correlation in our series of older patients, suggesting the possibility of a trade-off relationship between SL and SB. In the past, the trade-off structure of broadly defined SB versus LIPA and MVPA has been discussed in studies of physical activity mainly in community-dwelling older adults, and the re-placement of a large amount of SB time with LIPA and MVPA was proposed to improve health [32,33]. In a covariance study of community-dwelling older adults, SL and SB were shown to have the highest covariance, SB-LIPA and SB-MVPA had the lowest covariance, and SB was easily replaced with LIPA and MVPA [34]. Thus, the present results suggest differences in the temporal distribution of each physical activity intensity over the course of a day between hospitalized older adults and community-dwelling older adults.”

“Our present results indicate that in hospitalized older adults, SL and SB may have a conflicting relationship with the BBS score. The relationship that we observed between SB and physical function is similar to the relationship between physical function and broadly defined SB among community-dwelling older adults. On the other hand, the relationship between the amount of SL time and BBS is incongruous. Our present results are inconsistent with reports that prolonged bed rest leads to hospitalization-related dysfunction [5,6]. Since this pilot study was a cross-sectional univariate analysis, the effects of baseline differences and other confounding factors could not be excluded, and a longitudinal study or multiple analyses are thus necessary to clarify the causal relationships in this study. What is important to note here is that SL and SB may have different trends in their relationships with physical function.”

Round 2

Reviewer 1 Report

Thanks for the revisions and no further concerns.

Reviewer 2 Report

The revision of manuscript is accepted.